# Sulfaguanidine Hybrid with Some New Pyridine-2-One Derivatives: Design, Synthesis, and Antimicrobial Activity against Multidrug-Resistant Bacteria as Dual DNA Gyrase and DHFR Inhibitors

**DOI:** 10.3390/antibiotics10020162

**Published:** 2021-02-05

**Authors:** Ahmed Ragab, Sawsan A. Fouad, Ola A. Abu Ali, Entsar M. Ahmed, Abeer M. Ali, Ahmed A. Askar, Yousry A. Ammar

**Affiliations:** 1Department of Chemistry, Faculty of Science (Boys), Al-Azhar University, Nasr City, Cairo 11884, Egypt; 2Department of Chemistry, Faculty of Science (Girls), Al-Azhar University, Nasr City, Cairo 11754, Egypt; sawsan_ahmedfouad@yahoo.com (S.A.F.); drentsarmohamed74@gmail.com (E.M.A.); dr.abeer.mohamed.ali@gmail.com (A.M.A.); 3Department of Chemistry, College of Science, Taif University, P.O. Box 11099, Taif 21944, Saudi Arabia; O.abuali@tu.edu.sa; 4Department of Botany and Microbiology, Faculty of Science (Boys), Al-Azhar University, Nasr City, Cairo 11884, Egypt; drahmed_askar@azhar.edu.eg

**Keywords:** sulfaguanidine, antimicrobial activity, DNA gyrase and DHFR inhibitors, immunomodulatory potential, molecular docking study

## Abstract

Herein, a series of novel hybrid sulfaguanidine moieties, bearing 2-cyanoacrylamide **2a**–**d**, pyridine-2-one **3**–**10**, and 2-imino-2*H*-chromene-3-carboxamide **11**, **12** derivatives, were synthesized, and their structure confirmed by spectral data and elemental analysis. All the synthesized compounds showed moderate to good antimicrobial activity against eight pathogens. The most promising six derivatives, **2a**, **2b**, **2d**, **3a**, **8**, and **11**, revealed to be best in inhibiting bacterial and fungal growth, thus showing bactericidal and fungicidal activity. These derivatives exhibited moderate to potent inhibition against DNA gyrase and DHFR enzymes, with three derivatives **2d**, **3a**, and **2a** demonstrating inhibition of DNA gyrase, with IC_50_ values of 18.17–23.87 µM, and of DHFR, with IC_50_ values of 4.33–5.54 µM; their potency is near to that of the positive controls. Further, the six derivatives exhibited immunomodulatory potential and three derivatives, **2d**, **8**, and **11**, were selected for further study and displayed an increase in spleen and thymus weight and enhanced the activation of CD4^+^ and CD8^+^ T lymphocytes. Finally, molecular docking and some AMED studies were performed.

## 1. Introduction

The emergence and outbreak of methicillin-resistant *S. aureus* (MRSA), vancomycin-resistant Enterococci (VRE), and penicillin-resistant *S. pneumoniae* (PRSP) made antibiotic-resistant infections a critical healthcare concern worldwide [1]. A report published by the United Kingdom government predicted that antibiotic-resistant infections will result in 10 million deaths worldwide per year by 2050 if new antimicrobial strategies are not discovered [2]. Therefore, antimicrobial resistance is a significant public health threat. Although resistance occurs naturally, the overuse and misuse of broad-spectrum antibiotics has accelerated the increase in resistance [3]. Therefore, to avoid unintentional elicitation of antibiotic resistance or disruption of microbiota, the use of a species-selective antimicrobial agent, which specifically targets and kills the disease-causing strain, has been suggested [4,5,6]. However, no new class of antibiotics has been developed for infections triggered by Gram-negative bacteria, including *P. aeruginosa*, in the last 40 years; in addition, most of the antibiotics in late-stage clinical development belong to existing classes of drugs against which bacterial resistance has already been observed or could easily develop [7,8]. Thus, it is essential to modify and develop potent and effective antimicrobial agents to overcome the emerging multidrug-resistant strains of bacteria and fungi [9]. Sulfa drugs are the oldest chemically synthesized antimicrobial agents, as they still are widely used today for treatment of various bacterial, protozoal, and fungal infections [10]. Sulfonamides have diverse biological activities, including antibacterial activities (by inhibition of dihydropteroate synthase (DHPS), thus inhibiting the biosynthesis of dihydro folic acid) [11], carbonic anhydrase inhibitors (CA) [12], EGFR inhibitors [13,14], insulin-release inducers [15], as well as antiviral [16], antifungal [17], anticancer [18,19], and anti-inflammatory activities [20].

Guanidines, also called carbamidines, are a significant category of compounds that occur naturally or made artificially using different methodologies, and are important for their presence in some commercially available drugs [21,22]. Guanidine moieties represented as hydrophilic functional groups are also present in the side-chain arginine amino acid, which has been observed in various enzyme active sites and motifs of cell recognition as fumarate reductase, horseradish peroxidase, and creatine kinase [23]. It was reported that molecules that contain a guanidinium scaffold (cyclic or acyclic) are considered an important therapeutic agents due to its numerous pharmacological activities as anti-diabetics (metformin and phenformin) [24,25], antibiotics (chlorhexidine, streptomycin, and trimethoprim) [26,27], antihistamines (cimetidine) [28], and serine protease inhibitors (gabexate) [29] (Figure 1). Pyridine and its fused versions can be present in a broad variety of drugs, such as milrinone [30], which is useful for the treatment of the heart; Acetylcholine [31] enhancement is useful in the treatment of Alzheimer’s disease and 4-aminopyridine derivatives reported to have anti-amnesic activity [32].

Furthermore, 2-pyridones represent a unique class of pharmacophores that are represented in several antibiotics and therapeutic agents [33,34,35]. Additionally, 2-pyridones are a class of recently discovered potent antibacterial agents that are of interest due to their antibacterial potency against the bacterial type II DNA topoisomerases, which contain two homologous enzymes as topoisomerase IV and DNA gyrase [36]; also, 2-oxo-pyridine derivatives are isomeric with a 4-oxopyridine bioactive core that exist in the fluoroquinolone antibiotic class as ciprofloxacin, delafloxacin, and norfloxacin [37,38]. Moreover, cyanoacetamide derivatives are versatile reagents and possess nucleophilic and electrophilic centers [39], and therefore it can be represented as polyfunctional compounds that are used for synthesis of polyfunctionalized three-, five-, and six-membered rings and condensed heterocycles [40]. Many mechanisms were reported to be a significant target for antibacterial agents. One of these targets is the DNA gyrase enzyme found in all bacterial and controls, the topological state of DNA [41,42]. Similarly, the DHFR enzyme catalyzes NADPH reduction to NADP+ by converting dihydrofolate to tetrahydrofolate, which is an essential cofactor for the biosynthesis of purine nucleotides and thymidine, as well as many amino acids, and inhibition of this enzyme disrupts DNA synthesis and cell death [43,44]. The human immune system is a systematic defensive mechanism against various infectious organisms, such as viruses, fungi, bacteria, and parasites, and the immune response from these sources can cause disease [45,46]. The host immune response involves innate immunity and adaptive immunity. Both depend on some immune organs, such as the spleen, thymus, and immune cells (natural killer cells, macrophages, splenocytes, etc.), which represent role significantly in increasing the body’s immunity [47,48]. Immunomodulation, which is very essential for healthy growth, can change and maintain the level of the immune response to a desired level [49]. Activation of T lymphocytes, natural killer, and macrophage cells mainly produce the cytokines that have immunomodulatory potential and plays a vital role in the immune response process by assisting the expulsion of abnormal cells [50]. In view of the above observations and in continuation of our program on the medicinal chemistry that demonstrated biological activity [51,52,53,54,55,56,57,58,59,60]. We report herein the synthesis of the versatile hitherto unknown 2-pyridone and chromeno-[3,4-c] pyridine derivatives containing the sulfaguanidine moiety in a single molecular framework; the rational study design is presented in Figure 1. The newly synthesized guanidine derivatives were evaluated in vitro for their antibacterial and antifungal activities, as well as DNA gyrase and dihydrofolate reductase enzymes with immunomodulatory potential. Finally, molecular docking and some ADME studies were used to explain the obtained biological data.

## 2. Results and Discussion

### 2.1. Chemistry

The synthetic strategies adopted for constructing the target molecules are illustrated in Scheme 1, Scheme 2 and Scheme 3. The starting material, *N*-(4-(*N*-carbamimidoylsulfamoyl) phenyl)-2-cyano acetamide **1**, was prepared by cyanoacetylation of the sulfaguanidine with ethyl cyanoacetate in refluxing DMF. Elemental analysis and spectral data were in favor of the proposed structure. The IR spectrum of 2-cyanoacetamide derivative **1** showed absorption bands at *ʋ* 3441, 3336, 3232, 2264, and 1693 cm^−1^ due to the NH_2_, NH, CN, and CO groups. The ^1^H NMR spectrum showed a new singlet signal at *δ* 3.94 ppm corresponding to the methylene group, and a broad signal at *δ* 6.70 ppm due to four protons of the guanidine moiety. In addition, a singlet signal at *δ* 10.58 ppm was exchangeable with D_2_O for an NH proton, and two doublet signals in the region *δ* 7.65 and 7.72 ppm, with a coupling constant (*J* = 7.60 and 6.8 Hz) corresponding to the four aromatic protons. ^13^C NMR spectra showed two specific singlet signals, one at *δ* 27.38 ppm related to the methylene group and the second signal at *δ* 112.88 ppm for the cyano group, in addition to the presence of aromatic carbons in the region between *δ* 116.24 and 141.17 ppm, followed by two characteristic signals at *δ* 158.52 and 162.06 ppm for the C=N and C=O, respectively.

Knoevenagel condensation of the 2-cyanoacetamide derivative **1**, with aromatic aldehydes in the ethanolic piperidine under reflux conditions, furnished the corresponding acrylamide derivatives **2a**–**d** in good yield. The structure of acrylamide derivatives **2a**–**d** was an assignment based on their elemental and spectral data. The infrared spectrum of compound **2b** showed characteristic bands at *ʋ* 3360, 3309, 3206, 2222, and 1693 cm^−1^ assigned for the NH_2_, NH, CN, and C=O groups. The ^1^H NMR spectrum (DMSO-*d_6_*) of these compounds revealed two singlet signals at *δ* 3.89 and 8.25 ppm attributed to OCH_3_ and the methylinic proton (CH=). Furthermore, the four guanidine protons that appear as a broad signal at *δ* 6.74 ppm and four doublet signals between 7.19 and 8.04 ppm are attributed to eight aromatic protons, with a coupling constant (*J*) from 8.0 to 8.8 Hz. Its ^13^C NMR spectra afforded a new characteristic signal at *δ* 56.16 for the methoxy group and signals at *δ* 115.00, 158.56, and 163.35 ppm due to the CN, C=NH, and C=O carbons, respectively, beside signals between *δ* 115.42 and 151.36 ppm for the aromatic carbons.

Ternary condensation of cyanoacetamide derivative **1**, aromatic aldehyde, and malononitrile in an equimolar molar ratio (1:1:1) in basic ethanol (ethanol containing three drops of piperidine) afforded the 2-pyridone derivatives **3a**–**d**. The structure of the newly designed 2-oxopyridine derivatives **3a**–**d** was elucidated based on its analytical and spectral data. The structure of 2-oxopyridine derivatives **3a**–**d** was chemically confirmed by treating 2-cyanoacrylamide **2a**–**d** with malononitrile under reflux conditions and in the presence of piperidine as a catalyst (Scheme 1). Furthermore, the reaction of acrylamide derivative **2b** with 2-cyano-*N*-cyclohexyl acetamide in ethanolic piperidine afforded creating a product where its structure should be either **4** or **5**. The structure of the product was assigned as the pyrid-2-one derivative **4** rather than 2-amino-6-oxo-pyridin-3-carboxamide derivative **5**, where the formation of compound **4** is assumed to take place by an initial Michael addition of the active methylene to the double bond followed by cyclization through the addition of an NH of aliphatic acetamide to the cyano group. Finally, proton shifts occur to yield 2-aminopyridin-6-one derivative **4**.

Similarly, 2-amino-*N*-(4-(carbamimidoylsulfamoyl) phenyl)-5-cyano-4-(4-methoxyphenyl)-6-oxo-1,6-dihydropyridine-3-carboxamide **6** was prepared through the reaction of compound **2b** with cyanoacetamide in refluxing ethanol containing piperidine as catalyst (Scheme 2). The assignment of the structure of these derivatives was based on analytical and spectroscopic data. Thus, its IR spectrum of compound **6** displayed absorption bands at *ʋ* 3441, 3336, and 3232 cm^−1^ assignable to the NH_2_ and NH groups in addition to two bands at *ʋ* 2214 and 1689 cm^−1^ related to CN and carbonyl groups, respectively. Further, the ^1^H NMR spectrum of 6-aminopyridin-2-one derivative **6** was characterized by the existence of a methoxy group as a singlet signal at *δ* 3.89 in addition to the guanidine protons (NH_2_ + 2NH) at 6.71 ppm, as well as eight aromatic protons ranging between *δ* 7.20 and 8.50 ppm and two NH groups at *δ* 8.25 and 10.57 ppm. While, the ^1^H NMR spectrum of pyridine-2-one derivative **4** exhibited a three new singlet signals at *δ* 1.25, 2.75, and *δ* 2.90 ppm, attributed for five CH_2_ groups of the hexyl moiety. In the ^13^C NMR spectra of 6-aminopyridin-2-one derivative **4**, we observed new signals at *δ* 29.50, 31.25, and 36.28 ppm for the hexyl moiety, besides methoxy, CH-pyridine and two carbonyl groups, at *δ* 56.17, 62.59, 162.84, and 163.37 ppm, respectively.

The reactivity of cyanoacetamide derivative **1** towards some active methylene reagents was investigated. Thus, *N*-carbamimidoyl-4-(4,6-diamino-3-cyano-2-oxopyridin-1(2*H*)-yl) benzene-sulfonamide (**7**) was prepared by treating of cyanoacetamide derivative **1** with malononitrile in a molar ratio of 1:1. The structure of diamino-2-oxopyridine derivative **7** was confirmed based on its elemental analysis and spectral data. The formation of diamino pyridine derivative **7** is assumed to proceed via the Michael addition of cyanoacetamide derivative **1** to the cyano group of malononitrile to form the acyclic Michael adduct, followed by in situ cyclization to the pyridin-2-one skeleton. The IR spectra of pyridine derivative **7** revealed an abroad absorption band at *ʋ* 3410, 3332, and 3224 cm^−1^ besides bands at *ʋ* 2214 and 1651 cm^−1^ for the amino and imino (NH_2_ and NH), CN, and carbonyl groups, respectively. At the same time, ^1^H NMR spectra demonstrated three singlet signals at *δ* 4.13, 7.36, and 7.53 ppm, related to the CH- of pyridine and two amino groups at position four and six in the pyridine ring as well as both aromatic and guanidine protons. ^13^C NMR spectra also displayed signals at *δ* 71.37, 89.32, 114.98, 159.36, 170.80, and 162.02 ppm for CH-pyridine, the carbon of pyridine attached to the nitrile group, cyano, and the carbon-holding amino and carbonyl group, respectively.

Furthermore, cyclo-condensation of cyanoacetamide derivative **1** with acetylacetone furnished *N*-carbamimidoyl-4-(3-cyano-4,6-dimethyl-2-oxopyridin-1(2*H*)-yl) benzenesulfonamide **8** via intramolecular hetero cyclization by loss of water molecule. Analytical and spectroscopic data can provide a reaction product. The IR spectrum showed the appearance of absorption bands *ʋ* 3464, 3425, 3194, 2218, and 1651 cm^−1^, corresponding to the NH_2_, NH, CN, and CO groups, respectively. Its ^1^H NMR spectrum exhibited two additional singlet signals at *δ* 1.99 and 2.41 ppm, assignable to the protons of two methyl groups in the pyridine ring and a singlet signal at *δ* 6.51 ppm attributed to pyridine-H_5_; also, its ^13^C NMR afforded two additional signals at *δ* 21.16, 21.98 ppm due to two methyl carbons. Condensation of 4,6-diethyl-2-oxopyridine derivative **8** with aromatic aldehydes, such as *p*-chlorobenzaldehyde and *p*-anisaldehyde, afforded the corresponding styryl derivatives **9a**, **b** based on the elemental and spectral analyses. The ^1^H NMR spectrum of compound **9b** revealed the absence of a methyl signal at *δ* 1.99 ppm and the presence of five significant signals at *δ* 2.41, 3.84, 7.12, 7.26, and 6.49 ppm due to the methyl and methoxy protons and two singlet signals for the styryl moiety (2 CH=CH), as well as pyridine-*H*_5_. Furthermore, its ^13^C NMR exhibited four signals for the CH_3_, OCH_3_, CN, and carbonyl groups at *δ* 22.25, 55.88, 114.99, and 162.81 ppm, respectively (Scheme 3).

A Gewald reaction of the activated methyl pyridone derivative **8** with elemental sulfur in EtOH/DMF, containing triethylamine as a basic catalyst, led to the formation of a product that was formulated as 4-(3-amino-6-methyl-4-oxothieno[3,4-*c*] pyridin-5(4*H*)-yl)-*N*-carbamimidoylbenzenesulfon-amide **10**. The structure of the prepared compound was elucidated based on elemental analysis and spectral data. IR spectrum of thieno[3,4-*c*] pyridine derivative **10** displayed bands at *ʋ* 3433, 3286, 3232, and 1668 cm^−1^, related to the (NH_2_ + NH) and carbonyl groups, beside the absent band for the cyano group. The ^1^H NMR spectrum of compound **10** was characterized by the existence of a singlet signal for thiophene-H at *δ* 7.96 ppm in addition to pyridine-*H* and a methyl group at *δ* 6.50 and 2.41 ppm, respectively. Its ^13^C NMR showed signals at *δ* 21.15, 160.47, 160.90, and 162.83 ppm due to the methyl, carbon of imino (C=NH), carbonyl, and carbon attached to the amino and sulfur in the thiophene ring.

Furthermore, condensation of cyanoacetanilide derivative **1** with salicylaldehyde using ammonium acetate as a catalyst gave *N*-(4-(*N*-carbamimidoylsulfamoyl) phenyl)-2-imino-2*H*-chromene-3- carboxamide **11** on the bases of elemental analyses and spectral data. The resulting chromene derivative **11** has latent functional constituents, which have the potential for further chemical transformations that give new routes for the preparation of substituted, polycondensed chromene derivatives. The reaction of chromene **11** with malononitrile in refluxing DMF/ethanol containing a catalytic amount of piperidine furnishes the novel chromeno[3,4-*c*] pyridine derivative **12** in a good yield. The molecular structure of **12** was established through analytical and spectral data. Its infrared spectrum showed absorption bands at *ʋ* 3441, 3348, 3238, 3182, 2202, and 1662 cm^−1^ due to the NH_2_, NH, CN, and C=O functional groups, respectively; also, its ^1^H NMR and ^13^C NMR agree with the proposed structure (Scheme 3). The ^1^H NMR spectra of chromeno[3,4-*c*] pyridine derivative **12** exhibited the disappearance of two singlet signals for the CH-4 of chromene and the NH of amide, which appears as downfield of the chromene-3-carboxamide derivatives **11**. Besides, the appearance of new signals related to the amine group with guanidine protons was confirmed by increasing the integration of protons at *δ* 6.74 ppm to six instead of four.

### 2.2. Biological Activity

#### 2.2.1. Antimicrobial Activity with Structure Activity Relationship (SAR) Study

The newly designed and modified sulfaguanidine derivatives **2a**–**d**, **3a**–**d**, **4**, **6**, **7**, **8**, **9a, b**, **10**, **11**, and **12** were screened in vitro for their antimicrobial activity against six bacterial strains and two fungal pathogens. The antimicrobial activity results are described in Table 1, which were obtained by measuring the inhibition zone (mm) with the conventional paper disk diffusion method according to reported methods [61,62,63]. Tetracycline and amphotericin B were used as the positive controls, with a significantly sized inhibition zone against all the tested pathogens.

As represented in Table 1, there were some interesting SAR study observations, such as where the guanidine derivatives possessed antimicrobial activity varying from 12 ± 0.20 to 33 ± 0.53 mm against all the tested microorganisms. Firstly, the 2-cyanoacrylamide derivatives **2a**–**d** exhibited good to promising activity and the order of activity can be presented as follows: **2d** > **2a** > **2b** > **2c.** Additionally, for the antibacterial strains, it was found that 3-(3-hydroxy-4-methoxyphenyl)-acrylamide derivative **2d** showed higher activity than 3-(4-methoxyphenyl)acrylamide derivative **2b** and 3-(4-chlorophenyl)acrylamide derivative **2a**, with the inhibition zone ranging between 27 ± 0.73 to 33 ± 0.53, 23 ± 0.20 to 25 ± 0.87, and 21 ± 0.2 to 27 ± 0.5 mm, respectively. While, replacement of the methoxy group in acrylamide derivative **2b** with the hydroxyl group **2c** did not enhance the activity within the inhibition zone (IZ) (17 ± 0.24 to 21 ± 0.85 mm) compared to tetracycline (20 ± 0.50 to 25 ± 0.22 mm). 

Furthermore, the reaction of acrylamide derivative **2a**–**d** with malononitrile causes cyclization to produce 3,5-dicyano pyridine-2-one derivatives **3a**–**d**. It is interesting to note that the 3,5-dicyano pyridine-2-one with 4-chlorophenyl in position four of pyridine derivatives **3a** showed broad and higher activity than 3-(4-chlorophenyl) acrylamide derivative **2a** and higher than any 3,5-dicyano pyridine-2-one derivatives **3b**–**d**, with an inhibition zone (IZ) between 18 ± 0.16 and 31 ± 0.40 mm. Similarly, pyridine-3-carboxamide derivatives **4** and **6** showed moderate activity. However, the presence of an *N*-cyclohexyl group in position one increased the activity from 23 ± 0.22 to 25 ± 0.35 mm for the Gram-positive strains and from 23 ± 0.74 to 27 ± 0.30 mm for the Gram-negative strains. Besides, *N*-hydropyridine-3-carboxamide derivative **6** demonstrated an IZ from 16 ± 0.54 to 18 ± 0.12 mm and 12 ± 0.61 to 15 ± 0.96 mm respectively. *N*-cyclohexyl pyridine-3-carboxamide derivative **4** exhibited no activity to the *S. typhi* strain, while pyridine-3-carboxamide derivative **6** showed no growth against *E. faecalis* and *P. aeruginosa*. 

Our study extended to study the different substituents’ effect in positions four and six on pyridine-2-one derivatives **7** and **8** as the methyl and amino groups. From the obtained data, it is clear that 4,6-dimethyl pyridine-2-one derivative **8** advertised higher activity than diamino derivative **7**, with an inhibition zone (IZ) from 23 ± 0.14 to 27 ± 0.50 mm compared to tetracycline (25 ± 0.22 to 20 ± 0.5 mm). The reaction of the methyl group at position four of 4,6-dimethyl pyridine-2-one derivative **8** to form 4-(arylstyrenyl) pyridine-2-one derivative **9a**, **b**, or the condensed pyridine structure as theino[3,4-*c*] pyridine derivative **10**, did not lead to an improvement in antimicrobial activity but displayed moderate activity and less than the parent structure **8**. Replacement of the pyridine nucleus with the pyrane moiety in 2-imino chromen-3-carboamide derivative **11** displayed a significant antibacterial activity with an IZ of 28 ± 0.11, 24 ± 0.29, and 29 ±0.54 mm and 25 ± 0.43, 24 ± 0.13, and 18 ± 0.36 mm for the Gram-positive bacteria. Besides, for the Gram-negative bacterial strains used in this study, introducing the pyridine nucleus into the chromene derivative led to decreased activity, as observed in the activity of the chromeno[3,4-*c*] pyridine derivative **12**.

For the antifungal activity determined against yeast strains *C. albicans* and *F. oxysporum*, most of the synthetized compounds showed moderate to good activity, with six derivatives (**3b**, **3d**, **4**, **7**, **9a**, and **12**) that had no activity against *F. oxysporum.* Furthermore, six derivatives (**2a**, **2d**, **3a**, **3c**, **8**, and **11**) were the most active compounds and displayed a higher or equipotent inhibition zone that ranged between 22 ± 0.21 and 27 ± 0.5 mm for *C. albicans*, and exhibited an IZ from 18 ± 0.45 to 22 ± 0.11 mm for *F. oxysporum* in comparison to the amphotericin B as a positive control (22 ± 0.2 and 18 ± 0.32 mm) against the two fungal strain respectively.

Finally, it is noteworthy to mention that modification of sulfaguanidine with pyridine derivative (**3a**, **8)**, acrylamide derivatives (**2a**, **2b**, and **2d**), and chromene-3-carboxamide derivative **11** exhibited an improvement in antimicrobial activity, with a significant inhibition zone.

#### 2.2.2. Minimal Inhibitory/Bactericidal Concentrations (MIC) and (MBC)

Depending on the antimicrobial screening, six *N*-carbamimidoylsulfamoyl derivatives (**2a**, **2b**, **2d**, **3a**, **8**, and **11**) were evaluated for their MIC and MBC, as described in Table 2 and expressed in µM. The selected molecules exhibited excellent antibacterial activity against all the tested strains, where the most promising designed compounds displayed MIC values ranging between 4.69 and 156.47 µM and MBC values from 4.66 µM to 219.88 µM against Gram-positive bacteria, in comparison to the tetracycline MIC values (70.31–140.63 µM) and MBC values (91.40–210.9 µM). Additionally, the MBC/MIC ratio values of nearly 2 proved that these compounds showed bactericidal properties, according to the Clinical and Laboratory Standards Institute (CLSI) standards and reported studies [61,64,65].

Compound **2d**, having the 3-(3-hydroxy-4-methoxyphenyl) acrylamide derivative, presented the most active derivatives against *B. subtilis* and *E. coli*, with MIC and MBC values of 4.69 and 9.38 µM and 2.33 and 4.66 µM, respectively. Furthermore, 6-amino-4-(4-chlorophenyl)-3,5-dicyanopyridin-2-one derivative **3a** exhibited the second most promising derivatives against the *B. subtilis* and *E. coli* strains, with MIC values of 9.61 and 16.69 µM and MBC values of 19.60 and 30.02 µM, compared with tetracycline (MIC = 70.31 and 35.14 µM and MBC = 91.40 and 42.16 µM, respectively). Moreover, 4,6-dimethyl pyridine-2-one derivative **8** displayed the most active derivative against *S. aureus*, with MIC and MBC values of 5.64 and 10.71 µM, respectively. Furthermore, both 2-cyano acrylamide derivative **2d** and 3,5-dicyano pyridine-2-one derivative **3a** showed MIC values of 13.40 and 16.69 µM and MBC values of 13.40 and 33.38 µM, as well as all the tested derivatives that observed activity MIC values less than the standard positive control; also, 3,5-dicyano pyridine-2-one **3a** still had the most active derivative against *E. faecalis*, with a MIC value of 8.33 µM and MBC value of 15.83 µM, while tetracycline showed values of 140.6 and 210.9 µM for MIC and MBC, respectively, followed by 2-cyanoacrylamide derivative **2d** and then dimethyl pyridine-2-one derivative **8**.

For Gram-negative bacteria, the most active derivatives were found to be most effective against the three tested strains (*E. coli*, *P. aeruginosa*, and *S. typhi*), with MIC values ranging between 2.33 and 137.43 µM and MBC values between 4.66 and 171.89 µM, compared to those of tetracycline (MIC = 35.14–140.6 µM and MBC = 42.16–196.89 µM). Except for 2-cyano-3-(4-methoxyphenyl) acrylamide **2b** and 3-(4-chlorophenyl)-2-cyanoacrylamide **2a**, for which we observed higher MIC and/or MBC values against the tested strains.

Finally, regarding the most promising activity, derivatives **2a**, **2b**, **2d**, **3a**, **8**, and **11** were revealed to be the best in inhibition of the growth of bacteria, with low MIC and MBC values. These values showed that the sulfonamide derivatives exhibited bactericidal activity depending on the MBC/MIC ratio ranging between 1 and 2. Furthermore, it seems that the tested derivatives are the most potent against Gram-positive strains, and the order of antibacterial activity is as follows: **2d** > **3a** > **8** > **11** > **2b** > **2a**.

#### 2.2.3. Minimal Inhibitory/Fungicidal Concentrations (MIC) and (MFC)

Similarly, the minimal inhibitory concentration (MIC) and minimal fungicidal concentration (MFC) were determined against a panel of fungal strains in Table 3 and expressed in µM. Among the promising derivatives, only two derivatives, 3-(3-hydroxy-4-methoxyphenyl) acrylamide derivative **2d** and 3,5-dicyanopyridin-2-one derivative **3a**, appeared to be sensitive against both *C. albicans* and *F. oxysporum*, while the other four derivatives, **2a**, **2b**, **8**, and **11**, were the most resistant.

Compound **3a** displayed the most potency with MIC (16.69, 33.38 µM) and MFC (26.69, 56.74 µM) values nearly to the positive control amphotericin B (MIC = 16.81, 33.63 µM and MFC = 37.26, 70.62 µM). While, compound **2d** observed a slightly elevated MIC (18.80, 37.60 µM) and MFC (37.60, 66.84 µM) against *C. albicans* and *F. oxysporum*, respectively.

Finally, all the promising derivatives indicated moderate to good antifungal activity, especially 2-cyano acrylamide derivative **2d** and another 3,5-dicyanopyridin-2-one derivative **3a**, which appeared to be the two most active derivatives. Besides, all the newly designed compounds showed MFC/MIC ratios between 1 and 2, proving that these derivatives have fungicidal activity.

#### 2.2.4. Multidrug-Resistant Bacteria (MDRB) Study

The potential of the newly designed *N*-carbamimidoylsulfamoyl derivatives **2a**, **2b**, **2d**, **3a**, **8**, and **11** encouraged us to evaluate its activity against MDRB, namely *S. aureus ATCC 43300*, *S. aureus ATCC 33591*, *E. coli ATCC BAA-196*, and *P. aeruginosa ATCC BAA-2111*). As described in Table 4 and Table 5, the most active compounds exhibited a good inhibition zone (mm) with moderate to potent activity against the MDRB strains, showing MIC values ranging between 4.69 and 40.52 µM and MBC values between 8.19 and 81.08 µM compared to norfloxacin (MIC = 2.44–9.80 µM and MBC = 4.38–16.65 µM).

Among all the tested compounds, two derivatives, **2d** and **3a**, demonstrated the strongest inhibition activity, with MIC values of 4.69–9.38 µM and 8.33–16.69 µM, as well as MBC values of 9.38–18.77 µM and 14.23–33.38 µM, respectively. In the same way, compounds **2b**, **2d**, and **3a** were found to be best against *P. aeruginosa*, with MIC values (9.38, 9.38, and 9.48 µM) and MBC values (18.77, 18.77, and 14.23 µM) that are lower or slightly close to those of norfloxacin (MIC = 9.80 µM and MBC = 16.65 µM). Scientifically, 3-(substituted-aryl) acrylamide derivative **2b** and **2d** showed broad-spectrum activity against especially *E. coli* as compared to norfloxacin, with MIC values of 4.69, 4.69, and 4.91 µM and MBC values of 9.38, 9.38, and 8.32 µM), respectively. This activity might due to increasing electron contributing groups (OH, OMe) in the aryl group rather than another derivative from 2-cyanoacrylamide **2a**. Alternatively, an electron-withdrawing group, such as the chloro group in the phenyl ring and cyano in 6-amino-3,5-dicyanopyridin-2-one derivative **3a**, enhanced the antibacterial activity, which was still less than acrylamide derivative **2d** but higher than dimethylpyridine-2-one **8** and chromene-3-carboxamide derivative **11**. 

Finally, the most promising derivatives exhibited good activity against the MDRB strains used in this study, and according to the Clinical and Laboratory Standards Institute (CLSI) standards [66,67], these derivatives shown bactericidal potential.

#### 2.2.5. In Vitro *S. aureus* DNA Gyrase and *E. coli* DHFR Enzymatic Assay

The most promising derivatives containing *N*-carbamimidoylsulfamoyl derivatives **2a**, **2b**, **2d**, **3a**, **8**, and **11** were evaluated to explore the antibacterial mechanism of hybrids series for their inhibitory activity against in vitro *S. aureus* DNA gyrase and *E. coli* DHFR (dihydrofolate reductase) assays, with ciprofloxacin and trimethoprim as the positive control, respectively.

As described in Table 6 and Figure 2, in vitro *S. aureus* DNA gyrase inhibitory capacity expressed by (IC_50_ µM) was firstly assessed for the most promising guanidine derivatives and displayed moderate to potent inhibitory activity, with IC_50_ values ranging between 18.17± 1.18 and 39.41 ± 1.15 µM as compared to the positive control ciprofloxacin (IC_50_ = 26.32 ± 1.76 µM). Compound **2d**, having an acrylamide derivative containing in position three aryl groups as 3-hydroxy-4-methoxyphenyl, has shown excellent inhibitory activity and found to be the most promising derivative (IC_50_ = 18.17 ± 1.18 µM), while exchanging this aryl with other aryl groups as 4-chlorophenyl or 4-methoxyphenyl in **2a** and **2b** showed less activity (IC_50_ = 23.87± 1.22 and 29.14 ± 1.93 µM). Notably, 6-amino-4-(4-chlorophenyl)-3,5-dicyanopyridin-2-one derivative **3a** discovered the second inhibitory potency derivatives (IC_50_ = 21.97± 1.35 µM), with a nearly 1.19-fold higher value than for ciprofloxacin. Furthermore, 4,6-dimethyl pyridine-2-one (containing a methyl group in position four instead of an aryl as well as a methyl group rather than the amino group in position six) derivative **8** and 2-imino-2*H*-chromene-3-carboxamide (containing a chromone moiety instead of a pyridine nucleus) derivative **11** exhibited the least inhibition towards *S. aureus* DNA gyrase in comparison to the other tested derivatives. 

To explain the antimicrobial activity and investigate another mechanism, the most active derivatives were assayed against in vitro *E. coli* DHFR, with the IC_50_ values expressed in (µM). The target compounds **2a**, **2b**, **2d**, **3a**, **8**, and **11** displayed IC_50_ values in lower micromole between 4.33 ± 0.18 and 17.64 ± 0. 54 µM. Notably, both 2-cyanoacrylamide **2d** and 3,5-dicyanopyridin-2-one derivative **3a** showed the most promising derivatives, with inhibitory concentrations (IC_50_ µM) of 4.33 ± 0.18 and 5.44 ± 0.95 µM (1.19- and 0.94-fold increases), respectively. Trimethoprim was used as a standard DHFR inhibitor (IC_50_ values of 5.16 ± 0.12 µM). Further, the sensitivity of the most promising activity to *S. aureus* DNA gyrase and *E. coli* DHFR have an order **2d** > **3a** > **2a** >**2b** > **8** > **11**.

Finally, it can conclude that sulfaguanidine’s hybridization enhances the antimicrobial activity (MICs and MBCs values), with a lower micromole, especially in hybrids with some acrylamide or pyridin-2-one derivatives and those having a specific substituent, as shown in both **2d** and **3a**. Moreover, the most promising derivatives with the guanidine-sulfonyl moiety as a sidechain core indicated that they could inhibit bacterial growth through different mechanisms. Significantly, the antimicrobial activity depended on the nature of the design and the hybrid’s core, and the aromatic moiety.

#### 2.2.6. Immunomodulatory Activity

##### In Vitro Intracellular Killing Activities

One of the broad methods used for determining immune disorders for patients in hospitals is the NBT test to measure their immune responses. The reduction in NBT dye provides information about the phagocytic and intracellular killing functions of neutrophils, which are important for microbiocidal activity [68]. The most promising derivatives (**2a**, **2b**, **2d**, **3a**, **8**, and **11**) were evaluated against in vitro intracellular killing activities using a nitro blue tetrazolium (NBT) reduction assay and the obtained results expressed in percentage (%): increasing the percentage led to an improvement in the killing ability of the neutrophils used as innate immunity.

As represented in Table 7, intracellular killing activities displayed potency as an immunomodulatory agent with a percentage of 82.8 ± 0.37 to 142.4 ± 0.98. Remarkably, 3-(3-hydroxy-4-methoxyphenyl) acrylamide derivative **2d** showed the highest and most promising immunomodulatory action by killing activities (142.4 ± 0.98%), and the activity dramatically decrease in order of **8** > **11** > **3a** > **2b** > **2a**.

##### In Vivo Immunomodulatory Investigation

Depending on NBT assay, our work extended to study the effect of the most active three derivatives (**2d**, **8**, and **11**) on the in vivo immunomodulatory investigations and determining their effect on the immune organs, the thymus and spleen. Both the spleen and thymus have important antibacterial and fungal immune organ responses, and any change in weight, and indexes can reflect the level of immune regulation [47,69]. Additionally, the relative spleen and thymus indexes are important indices for nonspecific immunity and potential immunomodulatory compounds that increase the spleen and thymus weight. This increase is due to immune cells’ stimulation in the spleen and thymus [70,71,72].

The effect of 3-(3-hydroxy-4-methoxyphenyl) acrylamide derivative **2d**, 4,6-dimethyl pyridine-2-one derivative **8**, and 2-imino-2*H*-chromene-3-carboxamide derivative **11** on spleen and thymus weight and other indices are presented in Table 8, compared to the normal values and vitamin C as the positive control. Our study displayed that the promising three derivatives (**2d**, **8**, and **11**) exhibited a significant increase in spleen and thymus weight (*p* ≤ 0.05) compared to the normal control and nearly positive control. The 2-cyano-acrylamide derivative **2d** demonstrated an increase in spleen and thymus weights (250 ± 0.03, 15.56 ± 0.54 mg) compared to the normal (90 ± 0.02, 19.63 ± 0.35 mg) and nearly to vitamin C, with 260 ± 0.028 and 17.2 ± 0.31 mg, as well as slightly decreased spleen and thymus indices (0.018 ± 0.00 and 0.532 ± 0.06 mg/g) in comparison to the positive control (0.019 ± 0.00 and 0.633 ± 0.06 mg/g). Although, pyridine-2-one derivative **8** and 2-imino-2*H*-chromene-3-carboxamide derivative **11** showed good indexed results compared to the normal and positive control. Moreover, acrylamide derivative **2d** showed the highest spleen and thymus indexed values among the tested compounds. Finally, it can be concluded that the three derivatives have immunomodulatory potential.

Activation of immune cells as CD4^+^ and CD8^+^ T cells is one possible mechanism of a drug’s immune stimulus activity. An increase in these cells is thought to be a good indicator of an active immune response to infections [73]. So, our work extended to study the effect of the most promising derivatives (**2d**, **8**, and **11**) on T lymphocyte subsets (CD4^+^ and CD8^+^) from peripheral blood, and determined the relative level in model mice that analyzed by flow cytometry after the period of treating.

As represented in Table 9, there is diversity in the percentage of CD4^+^ and CD8^+^. 2-Cyano acrylamide derivative **2d**, having an aryl group with electron-rich hydroxy and methoxy groups, showed the highest percentage, 79.61 ± 0.3. While, 2-imino-2*H*-chromene-3-carboxamide derivative **11** exhibited the lowest values, 75. 20 ± 0.97%, compared to the positive control, with a 76.74 ± 0.7% ratio for the T lymphocyte subsets CD4^+^.

Similarly, the percentage of CD8^+^ T lymphocytes for the tested derivative ranged between 18.44 ± 0.2 and 27.05 ± 0.5. The acrylamide derivative **2d** displayed the highest percentage, 27.05 ± 0.5, and pyridine-2-one derivative **8** had the lowest value, 18.44 ± 0.2%, compared to the positive control (19.62 ± 0.21%). Finally, this increase and decrease in the percentage of the two T lymphocyte subsets suggested that the newly designed compounds **2d**, **8**, and **11** could enhance the stimulation and activation of CD4^+^ and CD8^+^ T lymphocytes, suggesting the T lymphocytes as a possible mechanism.

### 2.3. In Silico Studies

#### 2.3.1. Prediction of Some Physicochemical, Pharmacokinetic, and Toxicity Properties

Physicochemical properties are a widely recognized tool for designing bioavailable drugs and many efforts to assess drug "developability’’ depend on the calculated and measured physicochemical parameters [74]. As described in Table 10, some of the molecular properties for the most promising derivatives, **2a**, **2b**, **2d**, **3a**, **8**, and **11**, were calculated using the Swiss ADME tool (http://swissadme.ch/index.php (accessed on 21 October 2020)), as described in previously reported methods [59], to evaluate the drug-likeness properties. All the most active derivatives were found to have a number of rotatable bonds, a hydrogen bond acceptor and donor, as well as molecular weight and lipophilicity properties (MLogP), with the criteria limitation of Lipinski’s rule without any violation. These derivatives are suggested to have oral bioavailability properties. Similarly, three pharmacokinetic items were selected to be determined in our study, which are known as the blood–brain barrier (BBB) permeation, gastrointestinal absorption (GI), and permeability glycoprotein (P-gp). The designed compounds exhibited lower gastrointestinal absorption (GI) without blood–brain barrier (BBB) permeation and a non-substrate of the permeability glycoprotein (P-gp).

Furthermore, toxicity prediction is very important in the drug design process, because it makes it safer, more timely, cost less money, and reduces the number of animals for experimental purposes [75]. In our study, the most active six derivatives, **2a**, **2b**, **2d**, **3a**, **8**, and **11**, were evaluated against some toxicity items, such as the AMES toxicity, carcinogens, and acute oral toxicity category, using the admetSAR online program (http://lmmd.ecust.edu.cn/admetsar1/predict/ (accessed on 21 October 2020)). As represented in Table 11, the toxicity web tool admetSAR-predicted results, the promising derivatives revealed non-carcinogens and non-AMES toxicity, and all derivatives belonging to category III in acute oral toxicity observed LD_50_ values greater than 500 mg/kg but less than 5000 mg/kg; also, trimethoprim showed an acute oral toxicity category IV that displayed LD_50_ values greater than 5000 mg/kg. This web tool classified acute oral toxicity, depending on the LD_50_ values, into four categories based on the criterion of the US EPA. Further, ciprofloxacin demonstrated AMES toxicity, and both the standard drugs exhibited non-carcinogenic chemicals.

#### 2.3.2. Molecular Docking Studies

According to the antimicrobial results, the most active compounds (**2a**, **2b**, **2d**, **3a**, **8**, and **11**) were selected for molecular modeling simulation into DNA gyrase (PDB: 2XCT) and DHFR (PDB: IDLS) to investigate and understand the binding mode. As shown in Table 12, all docked compounds fitted well in the binding cavity of both the DNA gyrase and DHFR active sites and exhibited interaction, especially hydrogen bonding, arene–arene interactions, and arene–cation interactions. The molecular docking study was performed using Molecular Operating Environment software 10.2008 (MOE), Chemical Computing Group Inc., Montreal, Quebec, Canada.

As described in Table 12, the docking results observed a binding energy nearly or equal to the standard drugs. Firstly, analysis of the docking score energy (S) of the promising compounds inside the active site of DNA gyrase (PDB: 2XCT) exhibited that these derivatives had a comparable docking score energy, ranging between −15.98 and 21.67 Kcal/mol compared with ciprofloxacin (−9.87 Kcal/mol). The ciprofloxacin formed one hydrogen bond donor between the Thr580 and NH of the piperazine addition to the hydrogen bond acceptor between His1081 and oxygen of the carboxylate, with a bond length and strength of 2.6 °A (46%) and 2.3 °A (37%), respectively. The docking score of the most promising derivatives can be arranged in order **2d** > **3a** > **2a** > **8** > **2b** > **11** (see all figures of the docking study in the Appendix A).

For the guanidine derivative that has 3-(3-hydroxy-4-methoxyphenyl) acrylamide derivative **2d,** the lowest docking score energy was observed (S = −21.67 Kcal/mol), and considered the best compound in the active binding site of *S. aureus* DNA gyrase. This compound can form four side-chain hydrogen bond acceptors, or between His1079, Lys1043, Ser1173, and Gln1267 with the oxygen of sulfonyl, oxygen of acetamide, the phenolic hydroxy group of aldehyde and oxygen of methoxy group, respectively, with a bond length ranging between 2.5 and 2.8 °A and a strength varying from 24 to 66%; also, the phenyl of the aldehyde derivative formed an arene–cation with Arg1092 (Figure 3a).

Furthermore, 3,5-dicyano pyridine-2-one derivative **3a** revealed a binding energy (S = −20.78 Kcal/mol) with two critical hydrogen bond acceptors from side-chain between two oxygen of sulfonyl group and Arg1033 and His1079, with a bond length of 2.7 and 3.0 ˚A. Moreover, the amino of the guanidine interacted with Ser1085 through a hydrogen bond donor, with a bond length of 3.1 °A and a strength of 50%. Furthermore, the Arg1092 formed one hydrogen bond acceptor from side chain and one arene–cation with the cyano group in position three of the pyridine, with a bond length of 2.7 °A (50%) and phenyl group at position four of the pyridine, respectively (Figure 3b).

Similarly, the NH (imino) of the guanidine moiety at 2-cyanoacrylamide derivative **2a** formed two hydrogen bond acceptors with two NH of the residue Arg1092, with a bond length of 3.2 °A (12%) and 2.8 °A (12%) and binding energy of S = −20.27 Kcal/mol. Besides, two side-chain acceptors formed between the oxygen of sulfonyl and cyano of acrylamide, with His1046 and Arg1033 residues with bond lengths of 2.8 °A and 3.0 °A, respectively.

On the other hand, to explain the activity of the most active six derivatives on the DHFR enzyme, the docking simulation inside the active site of DHFR (PDB: IDLS) showed that the binding energy (S) ranged between −18.38 and −24.70 Kcal/mol compared with methotrexate (MTX, S = −24.44 Kcal/mol). The original ligand methotrexate (MTX) displayed nine hydrogen bonds, with different interactions as the hydrogen bond acceptor or donor, arene–arene, and hydrophobic interactions. Visual inspection of the docking poses of compounds **2a** and **3a** showed the same binding interaction by one hydrogen acceptor and another side-chain acceptor between residues Ser118 and Thr146 with two oxygens of the sulfonyl group with a slightly changed binding energy. The 2-cyanoacrylamide derivative **2a** displayed a binding energy of S = −20.26 Kcal/mol (Figure 3c). Furthermore, pyridine-2-one derivative **3a** revealed a docking score of S = −19.54 Kcal/mol, and the interaction is represented in (Figure 3d). Compound **2d** demonstrated the lowest binding energy, S = −24.70 Kcal/mol, with only one hydrogen bond acceptor between the Thr56 and NH of acetamide (Figure 3e). Similarly, 2-imino-2*H*-chromene-3-carboxamide derivative **11** exhibited one hydrogen bond acceptor between the oxygen of the sulfonyl and the residue Ser119, with a bond length of 2.7 °A and strength of 88%. Further, 4,6-dimethyl pyridine-2-one derivative **8** advertised two hydrogen bond donors between the residue Glu30 and two NH of hydrophilic functional groups with an equal bond length of 2.5 °A.

Finally, the docking results displayed that these new hybrid derivatives **2a**, **2b**, **2d**, **3a**, **8**, and **11** fitted well and interacted with the residues inside the active cavities in both DNA gyrase (PDB: 2XCT) and DHFR (PDB: IDLS), suggesting that they could potential be one of the antimicrobial agents (all the figures are in the Appendix A).

## 3. Materials and Methods 

### 3.1. Chemistry

Uncorrected melting points (MPs) of all the newly designed compounds were determined and recorded on a digital Gallen Kamp MFB-595 instrument using open capillaries and the recorded MPs were reported as such. A SHIMAZDU IR AFFINITY-I FTIR spectrophotometer was used for recording the IR spectra within the range of 400–4000 cm^−1^. A Bruker 400 MHz spectrometer was used to assessment of the ^1^H and ^13^C signals in the NMR spectra relative to (Me)_4_Si as an internal standard. Chemical shift values were reported in ppm units. The data were presented as follows: chemical shift, multiplicity (s = singlet, d = doublet, t = triplet, q = quartet, m = multiplet, br = broad, and app = apparent), coupling constant(s) in Hertz (Hz), and integration. Mass spectra were recorded on a Thermo Scientific ISQLT mass spectrometer at the Regional Center for Mycology and Biotechnology, Al-Azhar University. Elemental analyses were carried out at Micro Analytical Unit, Cairo University, Cairo, Egypt. The progress of the reaction was monitored by TLC using UV detection. The information of the chemical used in this study was listed and provided in a Appendix A.

*N-(4-(N-Carbamimidoylsulfamoyl) phenyl)-2-cyanoacetamide* (**1**)

A mixture of sulfaguanidine (2.14 g, 0.01 mol) and ethyl cyanoacetate (5 mL) in DMF (5 mL) was heated under reflux for 5 h. The obtained solid was collected by filtration, dried, and recrystallized from ethanol.

Pale yellow crystals; Yield 85%; mp 172–174 °C; IR (KBr): ν/cm^−1^: 3441, 3336, 3232 (2NH, NH_2_), 2264 (CN), 1693(CO); ^1^H NMR (400 MHz, DMSO-*d6*) *δ*/ppm 3.94 (s, 2H, CH_2_), 6.70 (s, 4H, 2NH+NH_2_, exchangeable by D_2_O), 7.65 (d, *J* = 7.6 Hz, 2H, Ar-H), 7.72 (d, *J* = 6.8 Hz, 2H, Ar-H), 10.58 (s, 1H, NH, exchangeable by D_2_O); ^13^C NMR (101 MHz, DMSO-*d6*) *δ* 27.38 (-CH_2_-), 112.88 (-CN), 116.24, 119.22, 127.23, 127.77, 139.90, 141.17 (Ar-Cs), 158.52 (C=NH), 162.06 (C=O); MS (EI, 70 eV): m/z (%) = 281 [M^+^] (17.94%), 267 (100%); Anal. Calcd for C_10_H_11_N_5_O_3_S (281.29): calcd.: C, 42.70; H, 3.94; N, 24.90%; found: C, 42.50; H, 3.99; N, 24.80%.

*N-(4-(N-Carbamimidoylsulfamoyl) phenyl)-3-(aryl)-2-cyano acrylamide* (**2a**–**d**)

A mixture of 2-cyanoacetanilide derivative **1** (2.81g, 0.01 mol) and substituted aldehyde, namely, *p*-chlorobenzaldehyde, *p*-anisaldehyde, *p*-hydroxybenzaldehyde, and vanillin, respectively (0.01 mol), in DMF/ethanol (1:1) in the presence of three drops of piperidine was refluxed for 6 h. The precipitate formed on hot was filtered and recrystallized from ethanol.

*N-(4-(N-Carbamimidoylsulfamoyl) phenyl)-3-(4-chlorophenyl)-2-cyanoacrylamide* (**2a**)

Orange crystals; Yield 65%; mp 250–252 °C; IR (KBr): ν/cm^−1^: 3447, 3340, 3232 (NH, NH_2_), 2202 (CN), 1687 (C=O); ^1^H NMR (400MHz, DMSO-*d_6_*) *δ*/ppm: 6.96 (s, 4H, 2NH+NH_2_, exchangeable by D_2_O), 7.19 (d, *J* = 8.0 Hz, 4H, Ar-H), 7.38 (d, *J* = 8.0 Hz, 4H, Ar-H), 7.41(s,1H, CH = methylinic), 10.16 (s, 1H, NH, exchangeable by D_2_O); ^13^C NMR (100 MHz, DMSO-*d6*): 79.38(-C-CN), 118.71 (-CN), 112.01, 116.59, 120.98, 121.14, 125.08, 129.09, 129.15, 129.39, 133.93, 134.29, 135.66, 143.79, 153.14 (Ar-Cs), 158.37(C=NH), 161.62 (C=O); MS (EI, 70 eV): m/z (%) = 403 [M^+^] (15.66%), 41 (100%); Anal. Calcd for C_17_H_14_ClN_5_O_3_S (403.84): calcd.: C, 50.56; H, 3.49; N, 17.34%; found: C, 50.30; H, 3.20; N, 17.55.

*N-(4-(N-Carbamimidoylsulfamoyl) phenyl)-2-cyano-3-(4-methoxyphenyl) acrylamide* (**2b**)

Yellowish orange crystals; Yield 65%; mp 244–246; IR (KBr): ν/cm^−1^: 3360, 3309, 3206 (2NH, NH_2_), 2222 (CN), 1693 (C=O); ^1^H NMR (400 MHz, DMSO-*d6*) *δ*/ppm 3.89 (s, 3H, OCH_3_), 6.74 (s, 4H, 2NH+NH_2_, exchangeable by D_2_O), 7.19 (d, *J* = 8.8 Hz, 2H, Ar-H), 7.76 (d, *J* = 8.0 Hz, 2H, Ar-H), 7.79 (d, *J* = 8.8 Hz, 2H, Ar-H), 8.04 (d, *J* = 8.8 Hz, 2H, Ar-H), 8.25 (s, 1H, CH-methylinic), 10.55 (s, 1H, NH, exchangeable by D_2_O); ^13^C NMR (100 MHz, DMSO-*d_6_*): 56.16 (OCH_3_), 79.64 (-C-CN), 115.00 (-CN), 115.42, 117.22, 120.45, 124.77, 126.97, 132.32, 133.19, (Ar-Cs), 140.20 (C-NH), 151.36 (C=C), 158.56, (C=NH), 161.82 (C-OMe), 163.35 (C=O); Anal. Calcd for C_18_H_17_N_5_O_4_S (399.43): calcd.: C,54.13; H, 4.29; N, 17.53; found: C, 54.30; H, 4.17; N, 17.75%.

*N-(4-(N-Carbamimidoylsulfamoyl) phenyl)-2-cyano-3-(4-hydroxyphenyl) acrylamide* (**2c**)

Reddish brown crystals; Yield 72%; mp 256–258 °C; IR (KBr): ν/cm^−1^: 3406, 3352, 3298, 3271 (OH, NH, NH_2_), 2202 (CN), 1681 (C=O); ^1^H NMR (400 MHz, DMSO-*d6*) *δ*/ppm 5.54 (br, 4H, 2NH + NH_2_, exchangeable by D_2_O), 6.71 (d, *J* = 8.0 Hz, 2H, Ar-H), 6.94 (d, *J* = 8.0 Hz, 1H, Ar-H), 7.39 (d, *J* = 8.0 Hz, 1H, Ar-H), 7.39 (d, *J* = 8.4 Hz, 1H, Ar-H), 7.62 (m, 1H, Ar-H), 7.70 (d, *J* = 8.0 Hz, 1H, Ar-H), 7.78 (d, *J* = 8.4 Hz, 1H, Ar-H), 7.92 (s, 1H, -CH=C methylinic), 9.36, 9.97 (2s, 2H, OH+NH exchangeable by D_2_O); ^13^C NMR (101 MHz, DMSO-*d6*) *δ* 98.42 (-C-CN), 117.32 (-CN), 118.95, 120.28, 124.19, 125.44, 126.80, 130.03, 132.03, 135.93, 139.49 (C-NH), 147.99 (C=C), 151.91 (C-OH), 158.41 (C=NH), 163.97 (C=O); Anal. Calcd for C_17_H_15_N_5_O_4_S (385.40): calcd.: C, 52.98; H,3.92; N, 18.17; found: C, 52.71; H, 3.72; N, 18.03%.

*N-(4-(N-Carbamimidoylsulfamoyl) phenyl)-2-cyano-3-(3-hydroxy-4-methoxyphenyl) acryl- amide* (**2d**)

Orange crystals; Yield 80%, mp 234–236 °C; IR (KBr): ν/cm^−1^: 3419, 3388, 3216 (OH, 2NH, NH_2_), 2219 (CN), 1675 (C=O); ^1^H NMR (400 MHz, DMSO-*d6*) *δ*/ppm 3.46 (s, 3H, OCH_3_), 4.91 (s, br, 4H, 2NH+NH_2_, exchangeable by D_2_O), 6.72 (s, 1H, Ar-H), 6.81 (d, *J* = 8.4 Hz, 2H, Ar-H), 7.72 (d, *J* = 8.4 Hz, 2H, Ar-H), 7.78 (s, 1H, NH exchangeable by D_2_O), 7.88 (d, *J* = 8.4 Hz, 2H, Ar-H), 8.10 (s, 1H, CH=C methylinic), 9.74 (s, 1H, OH exchangeable by D_2_O); ^13^C NMR (101 MHz, DMSO-*d6*) *δ* 56.52 (OCH_3_), 98.90 (C-CN), 117.86 (-CN), 118.13, 120.33, 121.00, 126.91, 127.22, 132.70, 134.10, 139.88, 141.59 (Ar-Cs), 151.59 (C-OH), 158.54 (C-OCH_3_), 162.51 (C=NH), 166.82 (C=O); Anal. Calcd for C_18_H_17_N_5_O_5_S (415.42): calcd.: C, 52.04; H, 4.12; N, 16.86; found: C, 52.21; H, 4.02; N, 16.53%.

*4-(6-Amino-4-(aryl)-3,5-dicyano-2-oxopyridin-1(2H)-yl)-N-carbamimidoylbenzenesulfonamides* (**3a**–**d**)

**Method A:** A mixture of 2-cyanoacrylamide derivatives **2a**–**d** (0.01 mol) and malononitrile (0.01 mol) in DMF/ethanol (1:1) and three drops of piperidine was heated under reflux for 5 h, cooled, and poured onto ice. The solid product obtained was collected by filtration, dried, and recrystallized from ethanol. 

**Method B:** A mixture of 2-cyanoacetamide derivatives **1** (2.81 g, 0.01 mol); and arylidene malononitrile derivatives (0.01 mol) in DMF/ethanol (1:1) in the presence of three drops of piperidine refluxed for 5 h, cooled, and poured onto ice. The solid product obtained was collected by filtration, dried, and recrystallized.

*4-(6-Amino-4-(4-chlorophenyl)-3,5-dicyano-2-oxopyridin-1(2H)-yl)-N-carbamimidoylbenzenesulfonamide***(3a**)

Brown crystals; Yield 72%, mp 272–274 °C; IR (KBr): ν/cm^−1^: 3448, 3344, 3205 (2NH+NH_2_), 2202 (CN), 1670 (C=O); ^1^H NMR (400 MHz, DMSO-*d6*) *δ*/ppm 6.80 (br, 4H, 4H, NH_2_+2NH, exchangeable by D_2_O), 7.39 (br, 2H, NH_2_ exchangeable by D_2_O), 7.53 (d, *J* = 8.0 Hz, 2H, Ar-H), 7.59 (d, *J* = 8.4 Hz, 2H, Ar-H), 7.69 (d, *J* = 8.4 Hz, 2H, Ar-H), 7.95 (d, *J* = 8.8 Hz, 2H, Ar-H); ^13^C NMR (101 MHz, DMSO-*d6*) *δ* 76.11 (C-CN), 116.35 (-2CN), 128.19, 128.77, 128.87, 129.37, 129.68, 130.40, 131.41, 131.67, 134.04, 135.66, 136.72, 146.30, 157.50, 158.64 (Ar-Cs), 159.99 (C=NH), 160.72 (N-C-NH_2_), 162.83 (C=O); Anal. Calcd for C_20_H_14_ClN_7_O_3_S (467.89): calcd.: C, 51.34; H, 3.02; N, 20.96; found: C, 51.23; H, 2.87; N, 20.74%.

*4-(6-Amino-3,5-dicyano-4-(4-methoxyphenyl)-2-oxopyridin-1(2H)-yl)-N-carbamimidoylbenzenesulfonamide* (**3b**)

Brown crystals; Yield 65%, mp 256–258 °C; IR (KBr): ν/cm^−1^: 3140, 3194, 3352, 3452 (2NH + 2NH_2_), 2218 (CN), 1678 (C=O); ^1^H NMR (400 MHz, DMSO-*d6*) *δ*/ppm 3.87 (s, 3H, OCH_3_), 6.78–6.97 (br, 4H, NH_2_+2NH, exchangeable by D_2_O), 7.15 (d, *J* = 7.2 Hz, 2H, Ar-H), 7.48–7.59 (m, 4H, Ar-H), 7.94 (d, *J* = 7.2 Hz, 2H, Ar-H), 8.05 (br, 2H, NH_2_, exchangeable by D_2_O); ^13^C NMR (101 MHz, DMSO-*d6*) *δ* 55.83 (OCH_3_), 75.88, 88.36 (2C-CN), 114.51, 114.64 (2CN), 116.41, 117.07, 126.99, 128.17, 129.71, 130.29, 131.05, 136.74, 146.31, 157.47, 158.64 (Ar-Cs), 160.12 (C=NH), 161.32 (C-NH_2_), 161.65 (C=O); MS (EI, 70 eV): m/z (%)= 463 [M+] (15.26%), 302 (100%); Anal. Calcd for C_21_H_17_N_7_O_4_S (463.47): calcd. C, 54.42; H, 3.70; N, 21.16; found: C, 54.27; H, 3.44; N, 21.02%. 

*4-(6-Amino-3,5-dicyano-4-(4-hydroxyphenyl)-2-oxopyridin-1(2H)-yl)-N-carbamimidoylbenzenesulfonamide* (**3c**)

Reddish brown crystals; Yield 65%, mp 265–268 °C; IR (KBr): ν/cm^−1^ 3464, 3425, 3352, 3194, (OH+NH+NH_2_), 2218 (CN), 1651 (C=O); ^1^H NMR (400 MHz, DMSO-*d6*) *δ*/ppm 5.13 (s, 2H, NH_2_, exchangeable by D_2_O). 6.81 (br, 4H, NH_2_ + 2NH, exchangeable by D_2_O), 6.95 (s, 2H, Ar-H), 7.41 (s, 2H, Ar-H), 7.54 (d, 2H, Ar-H), 7.94 (s, 2H, Ar-H), 10.09 (s, 1H, OH, exchangeable by D_2_O); ^13^C NMR (101 MHz, DMSO-*d6*) *δ* 75.77, 88.05 (C-CN), 115.59, 115.80 (-CN), 116.51, 117.18, 125.32, 128.15, 129.74, 130.39, 130.64, 136.77, 146.27, 157.45 (Ar-Cs), 158.62 (C=NH), 159.94 (C-OH), 160.17 (C-NH_2_), 161.95 (C=O); Anal. Calcd for C_20_H_15_N_7_O_4_S (449.45): calcd. C, 53.45; H, 3.36; N, 21.82; found: C, 53.30; H, 3.20; N, 21.72%.

*4-(6-Amino-3,5-dicyano-4-(3-hydroxy-4-methoxyphenyl)-2-oxopyridin-1(2H)-yl)-N-carbamimidoylbenzenesulfonamide* (**3d**)

Orange crystals; Yield 75%, mp 250–252 °C; IR (KBr): ν/cm^−1^ 3441, 3344, 3213 (OH+NH+NH_2_); 2206 (CN), 1661 (C=O); ^1^H NMR (400 MHz, DMSO-*d6*) *δ*/ppm 3.84 (s, 3H, OCH_3_), 6.78 (br, 4H, NH_2_ + 2NH, exchangeable by D_2_O), 6.96 (d, *J* = 8.4 Hz, 2H, Ar-H), 7.02 (d, 2H, Ar-H), 7.14 (s, 2H, NH_2_, exchangeable by D_2_O), 7.52 (s, 1H, Ar-H), 7.94 (d, *J* = 6.1 Hz, 2H, Ar-H), 9.72 (s, 1H, OH, exchangeable by D_2_O); ^13^C NMR (101 MHz, DMSO-*d6*) *δ* 56.23(OCH_3_), 75.81, 88.11(C-CN), 112.94, 115.89 (2CN), 117.21, 121.94, 125.50, 128.18, 129.72, 136.75, 146.30, 147.67, 149.23 (C=NH), 157.45 (C-NH_2_), 158.63 (C-OH), 160.16 (C=O); Anal. Calcd for C_21_H_17_N_7_O_5_S (479.47): calcd.: C, 52.61; H, 3.57; N, 20.45; found: C, 52.42; H, 3.42; N, 20.15%.

*2-Amino-N-(4-(N-carbamimidoylsulfamoyl) phenyl)-5-cyano-1-cyclohexyl-4-(4-methoxyphenyl)-6-oxo-1,6-dihydropyridine-3-carboxamide* (**4**)

A mixture of 2-cyanoacrylamide derivative **2b** (3.9 g, 0.01 mol) and 2-cyano-*N*-cyclohexyl acetamide (1.6 g, 0.01mol) in DMF/ethanol (1:1) in the presence of three drops of piperidine was heated under reflux for 3 h, cooled, and poured onto ice. The solid product obtained was collected by filtration, dried, and recrystallized from ethanol.

Pale yellow crystals; Yield 75%, mp 272–274 °C; IR (KBr): ν/cm^−1^ 3441, 3336, 3213 (NH+NH_2_), 2218 (CN), 2843, 2931 (CH, Aliph.), 1689, 1651 (2C=O); ^1^H NMR (400 MHz, DMSO-*d6*) *δ*/ppm 1.25 (s, 2H, CH_2_- cycohexyl), 2.75 (s, 4H, 2 CH_2_- cycohexyl), 2.90 (s, 4H, 2 CH_2_- cycohexyl), 3.74 (s, 3H, OCH_3_), 3.89 (s, 1H, -CH-cycohexyl), 6.74 (s, 4H, NH_2_+2NH, exchangeable by D_2_O), 6.95 (s, 2H, NH_2_, exchangeable by D_2_O), 7.20 (d, *J* = 6.8 Hz, 2H, Ar-H), 7.41 (d, *J* = 7.6 Hz, 1H, Ar-H), 7.62–7.69 (m, 2H, Ar-H), 7.79 (d, *J* = 6.0 Hz, 2H, Ar-H), 8.06 (d, *J* = 7.6 Hz, 1H, Ar-H), 10.60 (s, 1H, NH exchangeable by D_2_O); ^13^C NMR (101 MHz, DMSO-*d6*) *δ* 29.50, 31.25, 36.28 (-CH_2_), 56.17 (-OCH_3_), 62.59 (-CH), 103.77 (-C-CN), 112.82 (-CN), 114.38, 115.42, 117.19, 119.52, 120.45, 124.75, 126.74, 126.99, 127.75, 131.22, 133.21, 134.02, 140.21, 141.30, 151.39 (C-OCH_3_), 158.56 (C=NH), 161.81 (C-NH_2_), 162.84, 163.37 (2C=O); MS (EI, 70 eV): m/z (%) = 563 [M^+^] (8.99%), 108 (100%); Anal. Calcd for: C_27_H_29_N_7_O_5_S (563.63) calcd: C, 57.54; H, 5.19; N, 17.40; found C, 57.30; H, 5.02; N, 17.14.

*2-Amino-N-(4-(carbamimidoylsulfamoyl) phenyl)-5-cyano-4-(4-methoxyphenyl)-6-oxo-1,6-dihydropyridine-3-carboxamide* (**6**)

A mixture of 2-cyanoacrylamide derivative **2b** (3.9 g, 0.01 mol) and cyanoacetamide (0.84 g, 0.01 mol) in DMF/ethanol (1:1), including three drops of piperidine, was refluxed for 5 h, then cooled and poured onto ice. The solid product formed was collected by filtration, dried, and recrystallized from ethanol. 

Yellow crystals; Yield 75%, mp 265–268 °C; IR (KBr): ν/cm^−1^ 3441, 3336, 3232 (NH, NH_2_), 2214 (CN), 1689 (C=O); ^1^H NMR (400 MHz, DMSO-*d6*) *δ*/ppm3.89 (s, 3H, OCH_3_), 6.71 (br, 6H, 6H, 2NH_2 +_ 2NH, exchangeable by D_2_O), 7.20 (d, *J* = 6.4 Hz, 2H, Ar-H), 7.78 (m, 4H, Ar-H), 8.05 (d, *J* = 6.8 Hz, 2H, Ar-H), 8.25, 10.57 (2s, 2H, 2NH exchangeable by D_2_O); ^13^C NMR (101 MHz, DMSO-*d6*) *δ* 56.17 (-OCH_3_), 103.75 (C-CN), 115.43 (-CN), 120.43, 124.74, 126.99, 133.20, 140.19, 141.28 (Ar-Cs), 151.39 (C=NH), 158.53 (C-OMe), 161.66 (C-NH_2_), 163.29 (2C=O); Anal. Calcd. for: C_21_H_19_N_7_O_5_S (481.49) calcd: C, 52.39; H, 3.98; N, 20.36; found C, 52.11; H, 3.71; N, 20.05.

*N-Carbamimidoyl-4-(4,6-diamino-3-cyano-2-oxopyridin-1(2H)-yl) benzenesulfonamide* (**7**)

A mixture of 2-cyanoacetamide derivative **1** (2.81 g, 0.01 mol) and malononitrile (0.66 g, 0.01 mol) in DMF/ethanol (1:1) in the presence of three drops of piperidine was refluxed for 6 h, cooled, and poured onto ice. The solid product obtained was collected by filtration, dried and recrystallized from DMF.

Dark brown crystals; Yield 60%; mp 260 °C; IR (KBr): ν max/cm^−1^: 3410, 3332, 3224 (NH + NH_2_), 2214 (CN), 1651 (C=O); ^1^H NMR (400 MHz, DMSO-*d6*) *δ*/ppm 4.13 (s, 1H, CH-pyridine), 6.67 (br, 4H, NH_2_ + 2NH exchangeable by D_2_O), 7.36, 7.53 (2s, 4H, 2NH_2_ exchangeable by D_2_O), 7.68 (d, *J* = 6.8 Hz, 2H, Ar-H), 7.75 (d, *J* = 8.0 Hz, 2H, Ar-H), ^13^C NMR (101 MHz, DMSO *d6*) *δ* 71.37 (CH-pyridine), 89.32 (C-CN), 114.98 (CN), 116.93, 119.17, 125.69, 127.21, 131.88, 137.99 (Ar-Cs), 158.62 (C=NH), 159.36 (C-NH_2_), 162.02 (C=O), 170.80 (C-NH_2_); Anal. Calcd for C_13_H_13_N_7_O_3_S (347.35): calcd.: C, 44.95; H, 3.77; N, 28.23; found: C, 44.72; H, 3.90; N, 28.35.

*N-Carbamimidoyl-4-(3-cyano-4,6-dimethyl-2-oxopyridin-1(2H)-yl) benzenesulfonamide* (**8**)

To a solution of 2-cyanoacetamide derivative **1** (2.81 g, 0.01 mol) and acetylacetone (0.01 mol) in DMF/ethanol (1:1), three drops from piperidine was added. The solution was heated under reflux for 3 h, then left to cool. The solid product obtained was collected by filtration, dried and recrystallized from ethanol. 

Pale yellow crystals; Yield 70%; mp 270 °C; IR (KBr): ν max/cm^−1^: 3464, 3425, 3352, 3194 (NH + NH_2_), 2218 (CN), 1651 (C=O); ^1^H NMR (400 MHz, DMSO-*d6*) *δ*/ppm 1.99, 2.41 (2s, 6H, 2CH_3_), 6.51 (s, 1H, CH-pyridine), 6.83 (s, 4H, NH_2_ + 2NH exchangeable by D_2_O), 7.51 (d, *J* = 8.4 Hz, 2H, Ar-H), 7.93 (d, *J* = 8.0 Hz, 2H, Ar-H); ^13^C NMR (101 MHz, DMSO-*d6*) *δ* 21.16, 21.98 (2CH_3_), 100.46 (-CH), 109.59 (C-CN), 116.24 (-CN), 127.44, 129.04, 139.98, 145.64 (Ar-Cs), 152.30 (C-CH_3_), 158.67 (-C=NH), 160.46 (C=O); MS (EI, 70 eV): m/z (%) = 345 [M^+^] (58.61%), 116 (100%); Anal. Calcd for C_15_H_15_N_5_O_3_S (345.38): calcd.: C, 52.16; H, 4.38; N, 20.28; found: C, 52.02; H, 4.12; N, 20.45%.

*N-Carbamimidoyl-4-(3-cyano-4-(4-substitutedstyryl)-6-methyl-2-oxopyridin-1(2H)-yl) benzenesulfonamide* (**9a**,**b**).

A mixture of 4,6-diamino-3-cyano-2-oxo-pyridine derivative **8** (3.45 g, 0.01 mol), *p*-chloro-benzaldehyde or *p*-anisaldehyde (0.01mol) in DMF/Ethanol (1:1) in the presence of three drops of piperidine was heated under reflux for 3 h, cooled, and poured onto ice. The solid product obtained was collected by filtration, dried and recrystallized from ethanol.

*N-Carbamimidoyl-4-(4-(4-chlorostyryl)-3-cyano-6-methyl-2-oxopyridin-1(2H)-yl) benzene-sulfonamide* (**9a**)

Orange crystals; Yield 66%; mp up 300 °C; IR (KBr): ν max/cm^−1^: 3429, 3348, 3205 (NH+NH_2_), 2984, 2864 (CH-Alip.), 2218 (CN), 1681 (C=O); ^1^H NMR (400 MHz, DMSO-*d6*) *δ*/ppm 2.41 (s, 3H, CH_3_), 6.51 (s, 1H, CH-pyridine), 6.84 (s, 4H, NH_2_ + 2NH exchangeable by D_2_O), 7.07 (s, 1H, -CH=C), 7.34 (s, 1H, -CH=C), 7.50 (d, *J* = 8.0 Hz, 2H, Ar-H), 7.58 (d, *J* = 8.4 Hz, 2H, Ar-H),7.74–7.85 (m, 2H, Ar-H), 7.92 (d, *J* = 7.6 Hz, 2H, Ar-H); ^13^C NMR (101 MHz, DMSO-*d6*) *δ* 21.98 (CH_3_), 100.45 (-C-CN), 109.59 (-CH), 116.37 (CN), 127.44, 129.04, 129.48, 129.67, 130.12, 134.33, 136.89, 139.98, 145.63, 152.29 (N-C-CH_3_), 158.67 (-C=NH), 160.46 (C-C=C), 160.90 (C=O); MS (EI, 70 eV): m/z (%) = 467 [M^+^] (12.61%), 43 (100%); Anal. Calcd for C_22_H_18_ClN_5_O_3_S (467.93): calcd.: C, 56.47; H, 3.88; N, 14.97; found: C, 56.30; H, 3.73; N, 15.07.

*N-Carbamimidoyl-4-(3-cyano-4-(4-methoxystyryl)-6-methyl-2-oxopyridin-1(2H)-yl) benzene sulfonamide* (**9b**)

Yellow crystals; Yield 75%; mp up 300 °C; IR (KBr): ν max/cm^−1^: 3429, 3332, 3213 (NH+NH_2_), 2931, 2839 (CH-Alip.), 2214 (CN), 1691 (C=O); ^1^H NMR (400 MHz, DMSO-*d6*) *δ*/ppm 2.41 (s, 3H, CH_3_), 3.84 (s, 3H, -OCH_3_), 6.49 (s, 1H, CH-pyridine), 6.84 (s, 4H, NH_2_ + 2NH exchangeable by D_2_O), 6.95 (d, *J* = 6.4 Hz, 2H, Ar-H), 7.06–7.08 (m, 2H, Ar-H), 7.12 (s, 1H, -CH=C), 7.26 (s, 1H, -CH=C), 7.51–7.54 (m, 2H, Ar-H), 7.95 (d, *J* = 7.2 Hz, 2H, Ar-H), ^13^C NMR (101 MHz, DMSO-*d6*) *δ* 22.25 (CH_3_), 55.88 (-OCH_3_), 103.20 (-C-CN), 109.59 (-CH), 114.99 (CN), 115.11, 115.20, 119.51, 127.38, 128.00, 129.03, 129.16, 129.55, 130.24, 132.31, 140.56, 145.57, 151.46 (N-C-CH_3_), 154.81 (-C=NH), 158.70 (C-C=C), 161.56 (C-OCH_3_), 162.81 (C=O); Anal. Calcd for C_23_H_21_N_5_O_4_S (463.51): calcd.: C, 59.60; H, 4.57; N, 15.11; found: C, 59.75; H, 4.42; N, 15.23.

*4-(3-Amino-6-methyl-4-oxothieno[3,4-c] pyridin-5(4H)-yl)-N-carbamimidoylbenzenesulfon-amide* (**10**)

A solution of 4,6-diamino-3-cyano-pyridin-2-one derivative 8 (2.75 g, 0.01 mol) and elemental sulfur (0.32 g, 0.01 mol) in DMF/ethanol (1:1) in the presence of three drops of triethylamine was refluxed for 8 h. The obtained solid was crystallized from dioxane.

Grey powder; Yield 60%; Mp: ˃ 300 °C; IR (KBr): ν max/cm^−1^: 3433, 3286, 3232 (NH+NH_2_), 1668 (CO); ^1^H NMR (400 MHz, DMSO-*d6*) *δ*/ppm 2.41 (s, 3H, CH_3_), 6.50 (s, 1H, CH-pyridine), 6.82 (s, 6H, 2NH_2_ + 2NH exchangeable by D_2_O), 7.50 (d, *J* = 8.0 Hz, 2H, Ar-H), 7.92 (d, *J* = 8.0 Hz, 2H, Ar-H), 7.96 (s, 1H, CH-thieno); ^13^C NMR (101 MHz, DMSO-*d6*) *δ*/ppm 21.15 (CH_3_), 109.60 (-CH), 116.24, 127.44, 129.03, 139.98, 145.62, 152.29, 158.67, 160.47 (-C=NH), 160.90 (C=O), 162.83 (S-C-NH_2_); MS (EI, 70 eV): m/z (%) = 377 [M^+^] (20.20%), 61 (100%); Anal. Calcd for C_15_H_15_N_5_O_3_S_2_ (377.44): Calcd.: C, 47.73; H, 4.01; N, 18.56; found: C, 47.87; H, 4.21; N, 18.41.

*N-(4-(N-Carbamimidoylsulfamoyl) phenyl)-2-imino-2H-chromene-3-carboxamide* (**11**)

A mixture of 2-cyanoacetamide derivative **1** (2.81 g, 0.01 mol) and salicylaldehyde (0.01 mol) in DMF/ethanol (1:1) in the presence of ammonium acetate (0.2 g) was heated under reflux for 3 h and cooled. The solid product obtained was collected by filtration, dried, and recrystallized from ethanol. 

Yellow crystals; Yield 90%; Mp: 218–220 °C; IR (KBr): *ν* max/cm^−1^: 3479, 3429, 3356, 3224, (NH+NH_2_), 1670 (CO); ^1^H NMR (400 MHz, DMSO-*d6*) *δ*/ppm 6.56 (d, *J* = 8.4 Hz, 1H, Ar-H), 6.75 (s, 4H, NH_2_ + 2NH exchangeable by D_2_O), 7.00 (d, *J* = 8.4 Hz, 1H, Ar-H), 7.26 – 7.33 (m, 1H, Ar-H), 7.41 (d, *J* = 8.8 Hz, 1H, Ar-H), 7.51 (d, *J* = 8.4 Hz, 1H, Ar-H), 7.65 – 7.72 (m, 1H, Ar-H), 7.79 (m, 1H, Ar-H), 7.84 (d, *J* = 7.2 Hz, 1H, Ar-H), 8.59 (s, 1H, CH-chromone), 9.32, 10.89 (2s, 2H, NH exchangeable by D_2_O), ^13^C NMR (101 MHz, DMSO-*d6*) *δ* 112.81, 117.17, 122.04, 124.79, 127.38, 127.74, 129.68, 131.24, 133.08, 136.91, 140.98, 142.94, 148.16, 158.54 (C-O), 160.73, 165.33 (C=NH), 172.66 (C=O); MS (EI, 70 eV): m/z (%) = 385 [M^+^] (25.61%), 90 (100%); Anal. Calcd for C_17_H_15_N_5_O_4_S (385.40): calcd.: C, 52.98; H, 3.92; N, 18.17%; found: C, 53.09; H, 3.71; N, 18.25%.

*4-(2-Amino-1-cyano-5-imino-4-oxo-4H-chromeno[3,4-c] pyridin-3(5H)-yl)-N-cabamimidoylbenzenesulfonamide* (**12**)

A mixture of compound **11** (3.8 g, 0.01 mol) and malononitrile (0.66 g, 0.01 mol) in DMF/ethanol (1:1) in the presence of three drops of piperidine was refluxed for 5 h, cooled, and poured onto ice. The solid product obtained was collected by filtration, dried, and recrystallized from DMF. 

Brown crystals; Yield 60%; Mp: 266–268 °C, IR (KBr): *ν*/cm^−1^: 3441, 3348, 3238, 3182 (NH+NH_2_); 2202 (CN), 1662 (C=O); ^1^H NMR (400 MHz, DMSO- *d6*) *δ*/ppm 6.74 (s, 6H, 2NH_2_ + 2NH exchangeable by D_2_O), 7.00 (d, *J* = 7.4 Hz, 1H, Ar-H), 7.17 (d, *J* = 8.4 Hz, 1H, Ar-H), 7.30 (d, *J* = 7.5 Hz, 1H, Ar-H), 7.38 (d, *J* = 8.0 Hz, 1H, Ar-H), 7.47−7.62 (m, 2H, Ar-H), 7.64 -7.77 (m, 2H, Ar-H), 10.42 (s, 1H, NH exchangeable by D_2_O); MS (EI, 70 eV): m/z (%) = 449 [M^+^] (30.28%), 415 (100%); Anal. Calcd for: C_20_H_15_N_7_O_4_S (449.45) calcd.: C, 53.45; H, 3.36; N, 21.82; found C, 53.23; H, 3.25; N, 21.63.

### 3.2. In Vitro Antimicrobial Activity (See Appendix A)

#### 3.2.1. Bacterial and Fungal Strains Used in the Study

The preliminary antimicrobial activity of the newly modified sulfaguanidine derivatives **2a**–**d**, **3a**–**d**, **4**, **6**, **7**, **8**, **9a**–**b**, **10**, **11**, and **12**, as well as tetracycline and amphotericin B (a positive control), were evaluated against eight microbial pathogens. These pathogens were divided as three Gram-positive stains (*B. subtilis* ATCC 6633, *S. aureus* ATCC 29213 and *E. faecalis* ATCC 29212), three Gram-negative strains (*P. aeruginosa* ATCC 27853, *E. coli* ATCC 25922 and *S. typhi* ATCC 6539), and two fungal strains *F. oxysporum* (RCMB 008002) and *C. albicans* (ATCC 10231). The multidrug-resistant Gram-positive strains were *S. aureus* ATCC 43300, and *S. aureus* ATCC 33591. The multidrug-resistant Gram-negative strains were *E. coli ATCC BAA*-196 and *P. aeruginosa* ATCC BAA-2111. Norfloxacin was used as a positive control for the multidrug resistance study.

#### 3.2.2. Determination of Inhibition Zones (mm)

The antimicrobial activity performed at the bacteriology laboratory, Botany and Microbiology Department, Faculty of Science, Al-Azhar University, Cairo, Egypt. The inhibition zone (IZ) of the tested derivatives expressed as the diameter (mm) according to the agar plate diffusion method [36,40,61,76].

#### 3.2.3. Determination the Minimal Inhibition Concentration (MIC) and Minimum Bactericidal Concentration (MBC)

According to Clinical and Laboratory Standards Institute (CLSI) guidelines, the minimal inhibition concentration (MIC) of the most promising derivatives, depending on preliminary screening of **2a**, **2b**, **2d**, **3a**, **8**, and **11**, were dissolved and loaded on paper disks with different concentrations according to reported methods [61,67]. The experiment was repeating for three-times. The MBC assay was determined by plating 10 μL of culture volume from the MIC assay onto TSB agar plates and colony formation was examined after 24 h at 37 °C and according to the reported method [65]. MBC is defined as the lowest compound concentration resulting in a ≥ 3-log reduction in the number of colony-forming units (CFU) [77].

#### 3.2.4. Determination DNA Gyrase and DHFR Inhibitory Assay

The in-vitro enzymic assay for both *S. aureus* DNA gyrase and *E.coli* dihydrofolate reductase (DHFR) enzymes for the most promising derivatives with ciprofloxacin and trimethoprim as positive control was carried out in the confirmatory diagnostic unit, Vacsera, Egypt, according to the previously reported procedures and represented as IC_50_ values [62,78,79].

#### 3.2.5. Determination Immunomodulatory Activity

In vitro intracellular killing activities for the most promising six derivatives **2a**, **2b**, **2d**, **3a**, **8**, and **11** were performed according to reported methods [65,68,80]. While in vivo, the immunomodulatory aspects involved (relative immune organs weight and indexes as well as determining the T lymphocytes subsets (CD4^+^ and CD8^+^)) for the most active three derivatives, **2d**, **8**, and **11**, were evaluated according to reported methods [47,69].

#### 3.2.6. Molecular Docking Study

Docking simulations were performed using Molecular Operating Environment (MOE) software version 2008.10. The most active compounds, **2a**, **2b**, **2d**, **3a**, **8**, and **11**, were drawn by chem draw 2014, and then exported to MOE. Energy minimization using the MMFF94x force field was calculated for each molecule. The crystal structure of both dihydrofolate reductase enzyme (DHFR), which co-crystallized with methotrexate (MTX) and *S. aureus* DNA gyrase in complex with ciprofloxacin, were downloaded from the protein data bank (PDB ID: 1DLS, 2XCT) [81,82]. For DNA gyrase, the protein structure prepared by selecting the G chain only for molecular docking and other repeated chain was deleted as well as the water molecules and the trigonal matcher placement used for this study. Further, DHFR (1DLS) that contains only one chain and the alpha trigonal matcher placement methods were selected for the previous protein and the London dG scoring function was used in the docking protocol. The docking process methodology was first validated by redocking the original ligand that co-crystalized inside the active site, with deviation (RMSD) values of 1.12 and 2.6 Å for DNA gyrase and DHFR, respectively. The docking of new compounds was performed according to standard protocols and previously reported works [62,65].

#### 3.2.7. Ethics Statement for Both Animal Models and for Using Volunteer Blood Cells

All experimental protocols were approved and performed in compliance with the guide for the care and use of laboratory animals, published by the National Institutes of Health (USA), and performed according to Institutional guidelines. In addition, all volunteers gave their consent by signing a consent form before they participated.

## 4. Conclusions

In this study, eighteen compounds based on a sulfaguanidine moiety hybrid with acetamide, acrylamide, 2-oxopyridine, and chromene-3-carboxamide derivatives were synthetized and evaluated against several bacterial and fungal strains. The most promising compounds were **2a**, **2b**, **2d**, **3a**, **8**, and **11**, displaying low micromolar antibacterial activity with MIC values ranging between 2.33 and 156.47 µM and MBC values between 4.66 and 219.88 µM against the bacterial strains in comparison to the tetracycline MIC values (70.31–140.63 µM) and MBC values (91.40–210.9 µM). The promising derivatives indicated moderate to good antifungal activity against *C. albicans* and *F. oxysporum*, and thus have fungicidal properties. Furthermore, these derivatives showed moderate to potent activity against MDRB strains, with observed MIC values ranging between 4.69 and 40.52 µM and MBC values between 8.19 and 81.08 µM compared to norfloxacin (MIC = 2.44–9.80 µM) and (MB= 4.38–16.65 µM); therefore, these derivatives exhibited bactericidal potential. Additionally, the most promising **2a**, **2b**, **2d**, **3a**, **8**, and **11** derivatives showed good interaction with DNA gyrase and DHFR enzymes with IC_50_ values ranging between 18.17 and 39.41 µM and 4.33 and 17.64 µM, respectively, as compared to ciprofloxacin (26.32 ± 1.76 µM) and trimethoprim (5.16 ± 0.12 µM). Among them, three derivatives, **2d**, **3a** and **2a**, demonstrated IC_50_ values as potent, or nearly so, as the positive controls for both enzymes. The in vitro intracellular killing activities represented potency as an immunomodulatory agent with a percentage from 82.8 ± 0.37 to 142.4 ± 0.98. Similarly, the most active three derivative, **2d**, **8**, and **11**, exhibited a significant increase in spleen and thymus weight (*p* ≤ 0.05) and enhance the stimulation of the CD4^+^ and CD8^+^ T lymphocytes. Finally, it is interesting to mention that the molecular docking study matches with the experimental results. Besides, these derivatives obey Lipinski’s rule and revealed non-carcinogens, non-AMES toxicity, and all derivatives belong to category III regarding acute oral toxicity.

## Data Availability

Data are avilable as Appendix A.

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
