# Peer review of "Sulfaguanidine Hybrid with Some New Pyridine-2-One Derivatives: Design, Synthesis, and Antimicrobial Activity against Multidrug-Resistant Bacteria as Dual DNA Gyrase and DHFR Inhibitors"

_antibiotics, 2021, doi:10.3390/antibiotics10020162_

Round 1
Reviewer 1 Report
The manuscript reports the preparation of eighteen compounds designed by hybridization of a sulfaguanidine motif with acetamide, acrylamide, 2-oxopyridine, and chromene-3-carboxamide. The synthetic effort has been considerable and the products have been well characterized. The antibiotic activity of the compounds was evaluated against several bacterial and fungal strains showing moderate to good activity also against multi-drug resistance bacteria. These study were correlated with an investigation of the inhibitory activity of DNA gyrase and DHFR enzymes, with the study of the immunomodulatory effect, and with molecular docking studies. The results are interesting and the work is competently done. I recommend publication.
Author Response
Comments and Suggestions for Authors
The manuscript reports the preparation of eighteen compounds designed by hybridization of a sulfaguanidine motif with acetamide, acrylamide, 2-oxopyridine, and chromene-3-carboxamide. The synthetic effort has been considerable and the products have been well characterized. The antibiotic activity of the compounds was evaluated against several bacterial and fungal strains showing moderate to good activity also against multi-drug resistance bacteria. These study were correlated with an investigation of the inhibitory activity of DNA gyrase and DHFR enzymes, with the study of the immunomodulatory effect, and with molecular docking studies. The results are interesting and the work is competently done. I recommend publication.
Response: Many thanks for the respected reviewer for these kind and encouraging words.

Reviewer 2 Report
The manuscript entitled, "Sulfaguanidine hybrid with some new pyridine-2-one derivatives: design, synthesis, antimicrobial activity against multi-drug resistance bacteria as dual DNA gyrase and DHFR inhibitors" by Ragab et al. synthesized and characterized a series of hybrid sulfaguanidine derivatives. Authors characterized their molecules using a different characterization techniques including NMR, FTIR, and mass spectrometry. Synthesized molecules were evaluated for their biological activities using different in vitro systems and in mice models. Additionally molecular modeling was leveraged to interpret and develop structure-activity relationships. Overall, authors synthesized about one and half dozen compounds; manuscript looks interesting, however there are a few deficiencies that authors should address:
- Provide and interpret FTIR and mass spectrometry data for synthesized molecules
- all the raw data for biological activities, such as from disk diffusion experiments should be provided in supplementary materials
- Data for Immunomodulatory Activity leveraging human neutrophils is provided only for one healthy subjects. Authors should collect additional data in at least 3-5 healthy subjects and revise conclusions accordingly. Generally, there is huge inter- and intra- human variabilities in immune cell responses and data from a single volunteer is not enough to make appropriate conclusions
- Authors should also provided details of ethics committee approval(s) for experiments in animal models and for using volunteer blood cells
Author Response
Comments and Suggestions for Authors
The manuscript entitled, "Sulfaguanidine hybrid with some new pyridine-2-one derivatives: design, synthesis, antimicrobial activity against multi-drug resistance bacteria as dual DNA gyrase and DHFR inhibitors" by Ragab et al. synthesized and characterized a series of hybrid sulfaguanidine derivatives. Authors characterized their molecules using a different characterization techniques including NMR, FTIR, and mass spectrometry. Synthesized molecules were evaluated for their biological activities using different in vitro systems and in mice models. Additionally molecular modeling was leveraged to interpret and develop structure-activity relationships. Overall, authors synthesized about one and half dozen compounds; manuscript looks interesting, however there are a few deficiencies that authors should address:
Response:
Many thanks for kind and encouraging words, we have considered these valuable comments, the comments and our responses are summarized in the following points.
Query (1): Provide and interpret FTIR and mass spectrometry data for synthesized molecules.
Response: All the IR spectra for the newly designed compounds were now present in supplementary material file.
ـــــــــــــــــــــــــــــــــــــــــــــــــــــــــــــــــــــــــــــــــــــــــــــــــــــــــــــــــــــــــــــــــــــــــــــــــــــــــــــــــــــــــــــــــــــــــ
Query (2): all the raw data for biological activities, such as from disk diffusion experiments should be provided in supplementary materials.
Response: We provided them in supplementary material file.
ـــــــــــــــــــــــــــــــــــــــــــــــــــــــــــــــــــــــــــــــــــــــــــــــــــــــــــــــــــــــــــــــــــــــــــــــــــــــــــــــــــــــــــــــــــــــــ
Query (3): Data for Immunomodulatory Activity leveraging human neutrophils is provided only for one healthy subjects. Authors should collect additional data in at least 3-5 healthy subjects and revise conclusions accordingly. Generally, there is huge inter- and intra- human variabilities in immune cell responses and data from a single volunteer is not enough to make appropriate conclusions
Response: Dear respected reviewer, we are sorry for this conflict; where the isolation of neutrophils in this experiment was performed by collecting a sample of peripheral blood (15 mL) from three volunteers (5 mL blood from each one), and then the samples taken and collected in preservative-free heparin by adding 4.5% dextran B in saline (2 mL).
ـــــــــــــــــــــــــــــــــــــــــــــــــــــــــــــــــــــــــــــــــــــــــــــــــــــــــــــــــــــــــــــــــــــــــــــــــــــــــــــــــــــــــــــــــــــــــ
Query (4): Authors should also provided details of ethics committee approval(s) for experiments in animal models and for using volunteer blood cells
Response: We provided the ethics in both manuscript and supplementary material file.
Ethics statement for both animal models and for using volunteer blood cells:
All experimental protocols were approved and performed in compliance with the guide for the care and use of laboratory animals, published by the National Institutes of Health (USA), and performed according to Institutional guidelines. On the other hand, all volunteers gave their consent by signing a consent form before they participated.

Reviewer 3 Report
In this contribution by Ragab and co-workers, the authors design and synthesis series of new sulfaguanidine derivatives, and examine antimicrobial activity against multi-drug resistance bacteria as dual DNA gyrase and DHFR inhibitors. The results are interesting and potentially attractive to the readership of Antibiotics. Overall, I judge this should be published in Antibiotics, subject to some minor revision as below.
- The information of the chemical used in this study should be listed. If it is too long, such information could be moved to SI
- Line 118, ‘NH2 + 2NH’ should be changed to the right form.
- Please add error bar to the data in Table 2, 3, and 5.
- Figure 3, IC50 should be IC50.
- The resolution of figure 4 is not high enough for publishing.
- There are format issues with Refs: 1, 3, 45, and 62.
Author Response
Comments and Suggestions for Authors
In this contribution by Ragab and co-workers, the authors design and synthesis series of new sulfaguanidine derivatives, and examine antimicrobial activity against multi-drug resistance bacteria as dual DNA gyrase and DHFR inhibitors. The results are interesting and potentially attractive to the readership of Antibiotics. Overall, I judge this should be published in Antibiotics, subject to some minor revision as below.
Response:
Many thanks for kind and encouraging words, we have considered these valuable comments, the comments and our responses are summarized in the following points.
Query (1):
- The information of the chemical used in this study should be listed. If it is too long, such information could be moved to SI
Response: The information of the chemical used in this study was listed and provided in a supplementary material file.
ـــــــــــــــــــــــــــــــــــــــــــــــــــــــــــــــــــــــــــــــــــــــــــــــــــــــــــــــــــــــــــــــــــــــــــــــــــــــــــــــــــــــــــــــــــــــــ
Query (2):
- Line 118, ‘NH2 + 2NH’ should be changed to the right form.
Response: We deleted this part that may causes conflict, and retyping the sentence by using (a broad signal at δ 6.70 ppm due to four protons of guanidine moiety)
ـــــــــــــــــــــــــــــــــــــــــــــــــــــــــــــــــــــــــــــــــــــــــــــــــــــــــــــــــــــــــــــــــــــــــــــــــــــــــــــــــــــــــــــــــــ
Query (3):
- Please add error bar to the data in Table 2, 3, and 5.
Response: The standard errors were added for tables 2, 3, and 5.
ـــــــــــــــــــــــــــــــــــــــــــــــــــــــــــــــــــــــــــــــــــــــــــــــــــــــــــــــــــــــــــــــــــــــــــــــــــــــــــــــــــــــــــــــــــ
Query (4):
- Figure 3, IC50 should be IC50.
Response: We modified it.
ـــــــــــــــــــــــــــــــــــــــــــــــــــــــــــــــــــــــــــــــــــــــــــــــــــــــــــــــــــــــــــــــــــــــــــــــــــــــــــــــــــــــــــــــــــ
Query (5):
- The resolution of figure 4 is not high enough for publishing.
Response: We modified it and all figures for docking study present in full size in supplementary material file.
ـــــــــــــــــــــــــــــــــــــــــــــــــــــــــــــــــــــــــــــــــــــــــــــــــــــــــــــــــــــــــــــــــــــــــــــــــــــــــــــــــــــــــــــــــــ
Query (6):
- There are format issues with Refs: 1, 3, 45, and 62.
Response: We modified them according to authors guidelines of journal format.

Round 2
Reviewer 2 Report
Authors have addressed my concerns.